# Probabilistic Predictions with Federated Learning

**DOI:** 10.3390/e23010041

**Published:** 2020-12-30

**Authors:** Adam Thor Thorgeirsson, Frank Gauterin

**Affiliations:** 1Dr. Ing. h.c. F. Porsche AG, 71287 Weissach, Germany; 2Institute of Vehicle System Technology, Karlsruhe Institute of Technology, 76131 Karlsruhe, Germany; frank.gauterin@kit.edu

**Keywords:** probabilistic machine learning, federated learning, Bayesian deep learning, predictive uncertainty

## Abstract

Probabilistic predictions with machine learning are important in many applications. These are commonly done with Bayesian learning algorithms. However, Bayesian learning methods are computationally expensive in comparison with non-Bayesian methods. Furthermore, the data used to train these algorithms are often distributed over a large group of end devices. Federated learning can be applied in this setting in a communication-efficient and privacy-preserving manner but does not include predictive uncertainty. To represent predictive uncertainty in federated learning, our suggestion is to introduce uncertainty in the aggregation step of the algorithm by treating the set of local weights as a posterior distribution for the weights of the global model. We compare our approach to state-of-the-art Bayesian and non-Bayesian probabilistic learning algorithms. By applying proper scoring rules to evaluate the predictive distributions, we show that our approach can achieve similar performance as the benchmark would achieve in a non-distributed setting.

## 1. Introduction

Modern end devices generate large amounts of data, which enables the widespread, commercial application of machine learning (ML) algorithms. The data are distributed over a large group of end devices and are commonly transferred to a central server, where the learning of the ML models can be performed using the entire dataset. This poses two problems: transferring the data may lead to high communication costs and the privacy of the users may be compromised [1]. To counter these problems, ML can be performed on-device, so that the data are kept localized on the device and are not uploaded to a central server. The most prominent on-device ML methods are distributed learning [2,3], gossip learning [4] and federated learning [5]. In this work, we focus on the application of federated learning (FL) algorithms.

In FL, each end device learns from the local data, and a centralized server creates a global model by aggregating the model weights received from the devices at regular intervals. The global model is then sent back to the devices where the learning continues. FL is typically applied when a large dataset is desired, but sharing data between users is not possible or too expensive. In a distributed setting, data may not be independent and identically distributed (IID) and a robust model should take uncertainty into account. However, FL is not commonly applied to probabilistic models. In many applications, uncertainty of estimations or predictions can be significant. Bayesian deep learning (BDL) is commonly applied to account for uncertainty in neural networks (NNs) [6]. However, BDL methods are computationally expensive in comparison to non-Bayesian methods and hardware may as well be a limiting factor [7]. The inclusion of predictive uncertainty in distributed settings should therefore be addressed.

In this work, we apply FL to generate a probabilistic model. Inspired by related work on probabilistic predictions with NNs, we propose the learning of a probabilistic model through FL by introducing weight uncertainty in the aggregation step of the federated averaging (FedAvg) algorithm. In that way, the end devices can calculate probabilistic predictions but only have to learn conventional, deterministic models. This paper is organized as follows: In Section 2, we discuss probabilistic predictions with ML and give an overview of related work. In Section 3, we present our method, FedAvg-Gaussian (FedAG). In Section 4, we evaluate the performance of our method and compare the results to benchmarks from related literature. Finally, Section 5 gives concluding remarks and an outlook.

## 2. Related Work

A probabilistic prediction (or stochastic prediction) is when the prediction takes the form of a probability distribution, instead of a scalar value [8]. The application of ML in this topic is of significant relevance. Probabilistic predictions are commonly used in geology [9], electricity markets [10], urban water consumption [11], wind power [12], driver behavior [13], vehicle dynamics [14] and electric vehicle driving range applications [15]. Two prominent probabilistic prediction methods are BDL and ensemble methods, on both of which a summary was given by Ashuka et al. [16]. In BDL, the model parameters, e.g., weights w, are random variables represented by probability distributions p(w). With a dataset D, consisting of features x and target variable *y*, the posterior distribution for the model weights can be derived using the Bayes’ rule, which states that the posterior distribution is proportional to a prior probability p(w|α) multiplied with likelihood p(D|w,β)
(1)$$p\left(w|D,\alpha ,\beta \right)\propto p\left(w|\alpha \right)p\left(D|w,\beta \right)\;,$$
where α is a precision parameter for the prior distributions on weights w and β is a noise precision parameter. For simplicity, we refer to the weight posterior distribution as p(w|D). To make predictions for new, unseen data, the predictive distribution is obtained with
(2)p(y|x,D)=∫p(y|x,D,w)p(w|D)dw.
The exact computation of (Equation 1) and (Equation 2) is usually intractable due to the non-linearity of NNs [17]. The integration over the posterior is commonly approximated with Monte Carlo (MC) methods, such as Markov chain Monte Carlo (MCMC) or Hamiltonian Monte Carlo (HMC) [6]. Alternative approximation methods are extended variational inference [18] and cubature rules based on the unscented transformation [19].

A recent survey on BDL was given by Wang and Yeung [20]. Traditional Bayesian neural networks (BNNs) do not scale well and the posterior p(w|D) is usually difficult to calculate and sample from, but various approximation approaches have succeeded in creating probabilistic NNs. In variational inference (VI), the posterior p(w|D) is approximated with a well-defined distribution Q(w|D) and variational free energy F is minimized to minimize divergence between p(w|D) and Q(w|D) [21]. In Bayes by Backprop, the variational free energy is not minimized naïvely but approximately using gradient descent [22]. In probabilistic backpropagation (PBP), the posterior is determined with a calculation of a forward propagation of probabilities followed by a backwards calculation of gradients [23]. Gal and Ghahramani use dropout to achieve a mathematical equivalent of a Bayesian approximation without probabilistic weights [24]. Maddox et al. proposed SWA-Gaussian (SWAG), where an approximate posterior distribution over NN weights is determined by observing the stochastic gradient descent (SGD) trajectory during the learning process [25].

An established alternative to BDL is the use of ensembles to generate a probabilistic prediction. Therefore, multiple scalar predictions are combined to infer a probabilistic prediction. The predictions are either calculated with several different models or with a single model with varying initial conditions or input data. In a statistical post-processing of the ensemble predictions, a single probability density is derived [26]. A simple method is fitting a probability distribution to the predictions, e.g., a normal distribution $N(\mu ,\sigma^{2})$, by setting μ equal to the ensemble mean and σ to the ensemble standard deviation [27]. Further techniques exist, such as the ensemble model output statistics (EMOS) method [28], which is common in the atmospheric sciences [29]. Numerical models, mechanistic models and ML algorithms can all be used as individual predictors in the ensemble, but in this work, we focus on the application of ML algorithms.

Deep ensembles are ensembles of NNs where each of the NNs predicts the parameters of a predictive distribution, e.g., μ and σ, and the ensemble prediction is then a mixture of Gaussians [7]. Snapshot ensembles are generated by taking snapshots of NN weights at local minima during the training process, thus achieving an ensemble of NNs by training a single NN [30]. Fast geometric ensembling also trains a single NN and explores the weight space to find a set of diverse weight samples with minimal loss, thereby generating an ensemble of NNs [31]. Depth uncertainty networks are ensembles of sub-networks of increasing depth which share weights, thus needing only a single forward pass [32]. Ensembles of other ML algorithms also exist, e.g., gradient boosting (GB) ensembles [33]. Out of these ensemble methods, deep ensembles (DE) have recently shown the quite promising results in terms of prediction performance. The nature of DE has a certain resemblance to distributed methods, i.e., independent and parallel training of multiple NNs.

The learning of probabilistic ML models in a distributed and federated setting is the central challenge of our work. In partitioned variational inference (PVI), federated approximate learning of BNNs is presented [34]. Sharma et al. presented an extension of PVI including differential privacy [35]. However, probabilistic predictions of continuous variables are not implemented and we are therefore unable to use these methods as benchmarks. Concurrent to our work, several articles on probabilistic FL were published. Kassab and Simeone introduced distributed Stein variational gradient descent (DSVGD), where non-random and interacting particles represent the model global posterior. Iterative updates of the particles are performed on the devices by minimizing the global free energy [36]. Al-Shedivat et al. proposed federated posterior averaging (FedPA), where the devices use MCMC to infer approximations of the local posteriors, and the server computes an estimate of the model global posterior [37]. Zhang et al. used FedAvg with differential privacy to learn a Bayesian long short-term memory (LSTM) network, where Monte Carlo dropout is applied to compute probabilistic forecasts of solar irradiation [38]. In the next section, we propose our alternative method for the application of FL to probabilistic ML models.

## 3. Federated Learning with Predictive Uncertainty

Our proposed method, FedAvg-Gaussian (FedAG), builds on the federated averaging (FedAvg) algorithm [5]. In FedAvg, clients perform quick local updates on the weights, which are then aggregated in a central server. In turn, the aggregated weights are then returned to the clients for further learning. FedAvg does not consider predictive uncertainty. However, before the weights are aggregated, information on their distribution over the clients is known. Xiao et al. showed that during FL, client weights become increasingly correlated but not closer to each other in terms of distance metrics [39]. This fact may be a sign that the client weights are a good, approximate Bayesian marginalization, i.e., the weigths represent multiple *basins of attraction* in the posterior [40]. In our algorithm, this information is used to introduce weight uncertainty in the aggregation step of the FedAvg algorithm. Therefore, a probabilistic model is approximated by treating the set of local weights in the ensemble as an empirical posterior distribution for the weights of the global model. Using the probabilistic model, inference is performed by calculating predictive distributions for new, unseen data.

A pseudo-code for FedAG is shown in Algorithm 1. In the aggregation step, a probability distribution is fitted to the set of client weights. The choice of this distribution is arbitrary, but for simplicity, we consider normal distributions in this work. Hence, the posterior distributions are found by calculating the mean value $\mu_{w}$ and variance $\sigma_{w}^{2}$ of weights $w^{\left(k\right)}$. In turn, the posterior distributions p(w|D) are returned to the clients. The clients use the expected value, i.e., the mean value $\mu_{w}$ of the weight posterior distributions to further iterate local updates to the global model using their own data, but calculate probabilistic predictions with p(w|D).
**Algorithm 1** FedAvg-Gaussian (FedAG). *C* is the fraction of devices used in each round, *K* is the total number of devices, $D_{k}$ is the data observed by device *k*, *B* is the batch size, *E* is the number of local epochs, η is the learning rate and *l* is the squared loss function.

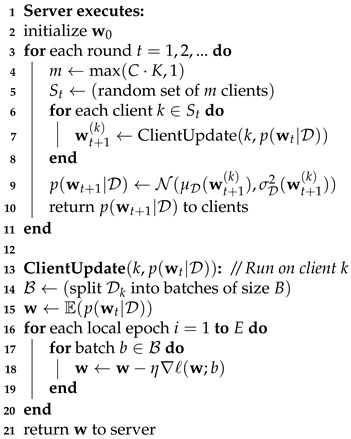


As in FedAvg, the clients minimize the mean squared error (MSE). The client updates are therefore fast and do not require extensions in order to learn a probabilistic model. FedAG is therefore significantly less complicated than PVI, DSVGD and FedPA, which is beneficial when resources such as computing performance and storage are limited. Additionally, the only target variable during training is μ, so that the amount of operations is smaller in comparison to DE. Figure 1 shows an overview of the training process where a network of end devices learns a probabilistic model. For $w_{i}$, the clients return their local updates, to which a normal distribution is fitted to generate a posterior probability distribution $p(w_{i}|D)$. The distributions $p(w_{i}|D)$ constitute the weights of the NN with input variable *x*, hidden units $H_{i}$, bias *I* and target variable $\hat{y}$. FedAG does not require prior probabilities on the weights.

As mentioned in Section 2, an exact calculation of the integral in (Equation 2) for the predictive distribution is generally intractable and some approximation is needed. We propose two variations for our algorithm: ordinary Monte Carlo (OMC) and non-parametric bootstrapping. In OMC, *M* sets of the weights are drawn from the posterior distributions to calculate *M* scalar predictions ${\hat{y}}_{k}$ for the target variable *y* [41]. It may seem strange to draw sets of sample weights from a distribution created by aggregating sets of sample weights. An alternative would be to use the sample weights from the clients directly to calculate the predictions. In a sense, this resembles non-parametric bootstrapping to create an ensemble [42]. In that way, ${\hat{y}}_{k}$ are calculated directly from the client updates. In this work, we will use this latter sampling method. The predictive distribution is approximated with a normal distribution $N({\hat{\mu }}_{y},{\hat{\sigma }}_{y}^{2})$: (3)$$\begin{array}{cc} p(y|x,D) & =\int p\left(y|x,D,w\right)p\left(w|D\right)dw:=N\left({\hat{\mu }}_{y},{\hat{\sigma }}_{y}^{2}\right) \end{array}$$(4)$$\begin{array}{cc} {\hat{\mu }}_{y} & \approx \frac{1}{M}\sum_{k=1}^{M}\hat{y}\left(x,w^{\left(k\right)}\right) \end{array}$$(5)$$\begin{array}{cc} {\hat{\sigma }}_{y}^{2} & \approx \left(\frac{1}{M}\sum_{k=1}^{M}{\left[\hat{y}\left(x,w^{\left(k\right)}\right)\right]}^{2}\right)-{\hat{\mu }}_{y}^{2}\;, \end{array}$$
where $\hat{y}\left(x,w^{\left(k\right)}\right)$ is a prediction calculated with features x and weights $w^{\left(k\right)}$. In the case of a linear model, the predictive distribution takes the form
(6)$$p\left(y|x,D,\alpha \right)=N\left(\mu_{w}^{T}x,\beta^{-1}+x\left[\sigma_{w}^{2}I\right]x^{T}\right)\;,$$
where β is a noise precision parameter for data D and is considered to be independent of the distribution of the weights w, I is the identity matrix, $\mu_{w}$ and $\sigma_{w}^{2}$ are the mean and variance of the weight posterior distribution p(w|D) [17].

## 4. Experimental Evaluation

As commonly done in the field of ML, we validate our proposed method with empirical data. In this section, we describe our experiments and analyze the results. In Section 4.1, we present a summary of proper scoring rules, which are necessary for the evaluation of probabilistic predictions. In Section 4.2, we apply FedAG to toy regression data. Section 4.3 shows the setup of the empirical validations and in Section 4.4, we present the results.

### 4.1. Proper Scoring Rules

To appropriately evaluate probabilistic predictions, proper scoring rules are needed. A scoring rule *S* is proper if
(7)$$E_{y∼P}\left[S(P,y)\right]\geq E_{y∼P}\left[S(Q,y)\right]\;,$$
where *P* is the true distribution of outcomes *y* and *Q* is the predictive distribution or any other probability distribution [43]. The scoring rule is strictly proper if the equality holds only when P=Q. Popular scoring rules for the prediction of continuous variables are the logarithmic score L(F,y), the continuous ranked probability score (CRPS) and its generalization, the energy score ES(F,y) [44]. In related work, negative log-likelihood (NLL) has been favored as a performance indicator. NLL is equal to the negative logarithmic score and is therefore also a proper scoring rule. Furthermore, NLL is unitless which is advantageous when evaluating a model’s performance on different datasets.

A good prediction is well calibrated and sharp. Calibration is the statistical consistency between the predictive distribution and the observation of the target variable. Sharpness measures the concentration of the predictive distribution. NLL measures both calibration and sharpness whereas root mean square error (RMSE) only measures calibration. Separate measures for calibration and sharpness allow a more detailed comparison. The width of a central prediction interval, e.g., 50%, was suggested by Gneiting and Raftery as a measure for sharpness [44]. As all candidate algorithms in this work calculate a prediction in the form of a normal distribution, the standard deviation appropriately measures the sharpness by indicating the width of the central 68% prediction interval. This is also called determinant sharpness (DS):(8)$$\mathrm{DS}=\det{\left(\Sigma \right)}^{1/2d}\;,$$
where $\Sigma \in R^{d\times d}$ is the covariance matrix of the predictive distribution and *d* is the dimension of the target variable. In our evaluation, we use the proper scoring rule NLL, as well as RMSE and DS.

### 4.2. Regression with Toy Data

To analyze the performance of our method on simple data, we generate a one-dimensional toy dataset as suggested by Hernández-Lobato and Adams [23]. In our analysis, 10 workers draw 16 independent examples from $y=x^{3}+\epsilon$ where $\epsilon ∼N(0,3^{2})$. Each worker trains a NN with a single hidden layer with 100 hidden units from these data according to Algorithm 1. The data are sampled in the interval $\left[-4,4\right]$ but predictions are calculated for the interval $\left[-6,6\right]$. Figure 2 shows the resulting probabilistic predictions after 1, 3 and 5 communication rounds. The results after t=5 rounds show that FedAG can calculate accurate probabilistic predictions with low but appropriate uncertainty for input data close to the observed training data. For input data farther away from observed data, the uncertainty is high. The prediction interval thus includes the ground truth, despite the scarce training data, making the prediction superior to those calculated after rounds t=1 and t=3. The predictions after t=5 rounds are similar to those reported by [7,23].

### 4.3. Experiment Setup

For the empirical validation, we implemented our method with two models: a NN with a single hidden layer and a linear regression model, denoted FedAG${}_{N=1}$ and FedAG${}_{N=0}$, respectively, where *N* represents the number of hidden layers. We use the experiment setup described by [23], which was also used by [7,24]. There, 10 datasets from the UCI Machine Learning Repository are used [45]. Table 1 shows a summary of the corresponding datasets. To ensure fair comparability to benchmark algorithms, the standard datasets are used.

We compare the performance of FedAG to three benchmarks: Bayesian linear regression (BLR) [17], variational inference (VI) [21], and deep ensembles (DE) [7], all of which are implemented in a non-distributed setting. We use Gaussian posteriors in VI and the DE consists of 5 networks. The NNs trained using VI, DE, and FedAG all have the same architecture with 50 hidden units with rectified linear unit (ReLU) activation functions in a single hidden layer. For the *Protein Structure* and *Year Prediction MSD* datasets, 100 hidden units are used. A 20-fold cross validation is performed to evaluate test performance, where E=40 passes over the available training data are done. For the *Protein Structure* dataset, a 5-fold cross validation is performed and for the *Year Prediction MSD* dataset, the specified split is used. The linear models, FedAG${}_{N=0}$ and BLR, are validated in the same manner. In the federated setting, K=10 devices are simulated with C=1 and batch size B=1. *K* and *C* are chosen so that the amount of data per device is maximized, subject to the condition that the number of devices is sufficiently large to enable an accurate approximation of the posterior distribution in the aggregation step. The training data are randomly divided into *K* equally large shards, each of which is assigned to a simulated device. Hence, each observation is uniquely assigned to one device.

The training of a linear model is a convex optimization and we expect that FedAG${}_{N=0}$ should need no more than t=1 rounds to converge. On the contrary, the training of a NN is usually a non-convex optimization and t>1 rounds are therefore required for convergence of FedAG${}_{N=1}$ in our setting. When each device has a limited amount of data, such as in small datasets or when the number of devices is increased, an even higher number of communication rounds might be required. Strong baselines in BDL are important and we try to generate a fair basis for the comparison of FedAG and the benchmarks [46]. For BLR and FedAG${}_{N=0}$, appropriate precision parameters for the variance of the target variable are estimated using the variance of the training data. In addition, conjugate priors given by unit Gaussians are used for the weight posterior distributions p(w|D) in BLR.

### 4.4. Results

With the experiment setup and proper scoring rules, we can evaluate the performance of FedAG and the benchmarks. In the following, we present the results of the validation. Table 2 shows the mean NLL and standard error for the algorithms on all dataset and Table 3 shows the RMSE and standard error. The results for VI and DE are reported by [7,23], respectively. The entries in bold denote the best performing model(s), where the performance is considered similar if the standard error intervals overlap.

The performance of the two linear models, BLR and FedAG${}_{N=0}$ is similar. In 8 out of 10 datasets, FedAG${}_{N=0}^{\left(t=1\right)}$ performs similarly or slightly better than BLR in terms of NLL. BLR significantly outperforms FedAG${}_{N=0}^{\left(t=1\right)}$ only in two datasets, the *Energy Efficiency* and *Yacht Hydrodynamics* datasets, which are also two of the smallest datasets. Hence, each worker only has access to a small amount of data. In 9 out of 10 datasets, the performance of FedAG${}_{N=0}^{\left(t=1\right)}$ and BLR is almost identical in terms of RMSE.

In the results for NNs, the difference between the three algorithms, VI, FedAG${}_{N=1}$ and DE is somewhat significant. DE achieve the best results, followed by FedAG${}_{N=1}^{\left(t=5\right)}$ and VI. In 8 out of 10 datasets, FedAG${}_{N=1}^{\left(t=5\right)}$ outperforms VI in terms of NLL and in 3 out of 10 datasets, the performance of FedAG${}_{N=1}^{\left(t=5\right)}$ approaches that of DE. In terms of RMSE, the performance of FedAG${}_{N=1}$ and DE is similar in 5 out of 10 datasets. Further rounds (t>5) do not improve the results of FedAG${}_{N=1}$ significantly.

To further compare the performance of VI, DE and FedAG${}_{N=1}$ over the course of the communication rounds, we look at the dataset *Concrete Compression Strength*, where the performance of the methods is similar, and the dataset *Yacht Hydrodynamics* where DE show a significant advantage in terms of NLL. Figure 3 shows the prediction performance (NLL and RMSE) of the algorithms on these two datasets. In Figure 3a,b, FedAG${}_{N=1}$ outperforms VI already after t=1 rounds, both in terms of NLL and RMSE. However, FedAG${}_{N=1}$ reaches a certain saturation and cannot match the performance of DE, despite a significant improvement in NLL and RMSE after t=5 rounds. In Figure 3c,d, FedAG${}_{N=1}$ and VI show similar performance after t=1 rounds. With increasing number of communication rounds *t*, NLL and RMSE of FedAG improve. After t=5 rounds, the results of FedAG${}_{N=1}$ and DE overlap, i.e., the algorithms achieve similar performance, though DE still retains a slight advantage. In the initial round of FedAG, different devices might find weights corresponding to different minima of the NN’s loss function, so that the global model’s initial weight posterior distributions are not optimal. The loss function of a NN with ReLU activation functions can be expressed as a polynomial function of the weights in the network, whose degree is the number of layers, and whose number of monomials is the number of paths from inputs to output [47]. We can therefore expect that loss functions of small NNs have few local minima, and that the global minimum can be found within relatively few communication rounds in FedAG. Accordingly, larger NNs might require more communication rounds. On the two datasets in Figure 3, we observe how the performance of FedAG${}_{N=1}$ gradually improves with increased rounds *t*. This can also be observed on other datasets.

Another important property of the predictive distributions is their sharpness, which we measure with determinant sharpness (DS). In Table 4, the mean DS of the predictive distributions calculated with FedAG${}_{N=1}^{\left(t=5\right)}$ and DE are shown. Of the datasets that exhibit similar performance in terms of NLL, *Boston Housing* and *Concrete Compression Strength* can be predicted with greater sharpness by FedAG than DE, whereas DE’s predictions of *Red Wine Quality* are sharper on average. In 7 out of 10 datasets, FedAG predicts on average a sharper distribution than DE.

### 4.5. Computational and Communication Complexity

In addition to the predictive performance of the algorithms, their computational and communication complexity is of significant importance. The candidate algorithms have different computational complexity at training time and at testing time. The two linear models, BLR and FedAG${}_{N=0}$, have the same structure and the same amount of parameters. The predictive distributions can be computed analytically with (Equation 6) and no sampling is required. The NNs trained using VI, DE, and FedAG are, on the other hand, more complex. VI and FedAG learn probabilistic NNs with Gaussian posterior, whereas DE are ensembles consisting of deterministic NNs with scalar weights, but with two output variables. VI maximizes a lower bound on the marginal likelihood of the NN. First, a Monte Carlo approximation for the lower bound is computed, which is then optimized using SGD. The computational complexity at training time is therefore higher in VI than in DE and FedAG, where SGD is applied directly. VI and FedAG approximate predictive distributions using Monte Carlo sampling from the posterior distributions. Contrarily, DE only have to analytically compute the two output variables of the 5 networks in the ensemble. Subsequently, the predictive distribution is approximated as a mixture of the individually computed normal distributions. The computational complexity at testing time is therefore higher in VI and FedAG than in DE.

The communication complexity of FedAG is different from that of FedAvg. FedAG learns a posterior distribution for each weight of the model. Therefore, its communication complexity is somewhat higher than that of FedAvg. If a Gaussian posterior is assumed, each distribution is defined by its mean and standard deviation. Compared to FedAvg, the global model has twice the amount of parameters. A single parameter can be assumed to be a 32 bit floating-point value. For the NNs considered in this work (single hidden layer, 6–90 features, 50 or 100 hidden units), the total data size is in the range from 1604 B to 36,804 B in FedAvg and from 3208 B to 73,608 B in FedAG. The communication complexity of sending the global model to the clients in FedAG can be up to two times higher than in FedAvg, depending on the communication overhead. However, the client updates only include scalar weights w, so the upload communication complexity in FedAG is the same as in FedAvg.

### 4.6. Discussion

As each of the clients in FedAG only has access to a fraction of the dataset, we do not expect it to out-perform the benchmarks BLR, VI and DE, which simultaneously have access to the complete dataset. Nevertheless, the linear models BLR and FedAG${}_{N=0}$ attain almost identical performance. Consequently, FedAG${}_{N=0}$ can be applied as an alternative to BLR in federated, distributed settings. In the case of a non-linear model, the performance of FedAG${}_{N=1}$ can generally compete with that of VI and approaches the performance of DE on some datasets. Additionally, the sharpness of the predictive distributions calculated with FedAG and DE is comparable. Hence, FedAG${}_{N=1}$ can be used as a probabilistic model in a federated setting, achieving predictive performance comparable with state-of-the-art non-federated and non-distributed methods. Further advantages of FedAG are the retained privacy and communication efficiency [48]. Therefore, FedAG offers prediction performance comparable with state-of-the art probabilistic ML algorithms in an efficient and privacy-preserving manner.

## 5. Conclusions and Future Work

Interest in the application of ML algorithms in distributed or federated settings is increasing. There, predictive uncertainty is an important feature that needs to be addressed. We presented FedAvg-Gaussian (FedAG), an efficient method for the learning of probabilistic models in a distributed, federated setting. FedAG extends FedAvg to include predictive uncertainty, by treating the set of local weights as a posterior distribution for the weights of the global model. Therefore, predictive uncertainty can be represented in a computation- and communication-efficient way, so that probabilistic on-device machine learning is realized.

We used FedAG to learn two different models, a linear regression and a feed-forward neural network with a single hidden layer. The performance of our method was evaluated on UCI regression datasets and compared to benchmark methods using proper scoring rules. When implemented with a linear regression model, FedAG’s performance is similar to that of a BLR. FedAG with a neural network can after t=5 communication rounds outperform VI on most datasets and its performance approaches that of DE on several datasets.

Our future work includes several topics. FedAG could benefit from an aggregation step more robust to outliers and adversarial attacks, such as in the methods Krum [49] and Aggregathor [50]. Moreover, further work should aim to test the methods with real federated data to analyze the effect of non-IID partitioning and stratified splits compared to randomized splits [51]. For the personalization of the local models, different aggregation and initialization methods can be applied. Evaluating such concepts in an asynchronous federated learning environment might prove an important area for future research. Furthermore, larger neural networks and other deep learning architectures such as convolutional neural networks can be applied and analyzed. Finally, our future research will aim to benchmark FedAG against other novel federated learning concepts, such as partitioned variational inference (PVI) [34], distributed Stein variational gradient descent (DSVGD) [36], federated posterior averaging (FedPA) [37], and the combination of FedAvg and Monte Carlo dropout [38].

## Figures and Tables

**Figure 1 entropy-23-00041-f001:**
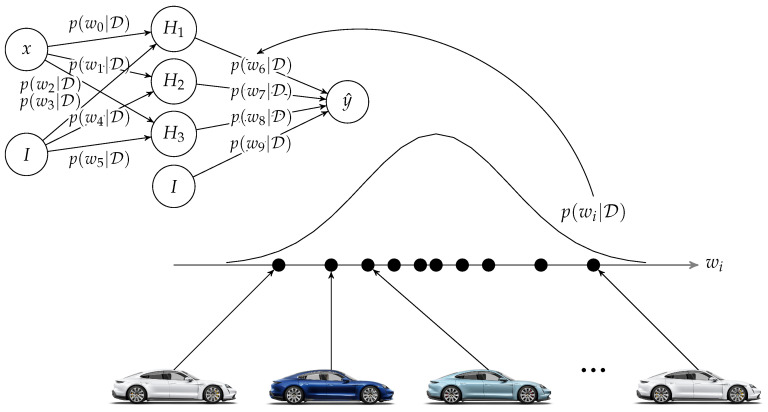
A network of end devices learns a probabilistic model.

**Figure 2 entropy-23-00041-f002:**
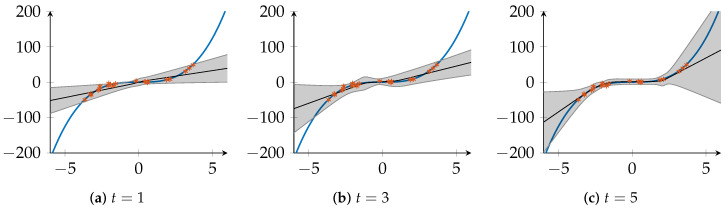
Results on regression for toy data after 1, 3 and 5 rounds. The blue line represents the ground truth, the orange points are exemplary observed noisy training data, the black line is the mean value of the predictive distribution and the grey area demarcates a prediction interval containing ±3 standard deviations.

**Figure 3 entropy-23-00041-f003:**
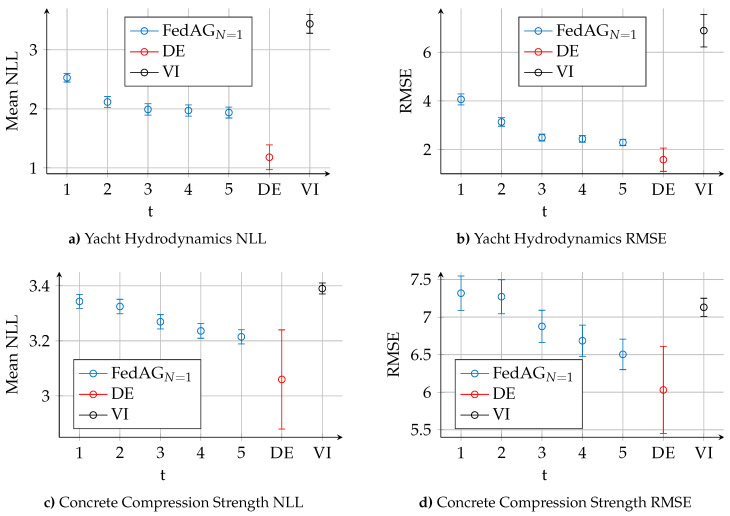
Prediction performance of FedAG${}_{N=1}$, DE and VI in terms of NLL and RMSE on the datasets *Yacht Hydrodynamics* and *Concrete Compression Strength*.

**Table 1 entropy-23-00041-t001:** Summary of the UCI datasets for regression.

Datasets	Observations	Features
Boston Housing	506	13
Concrete Compression Strength	1030	8
Energy Efficiency	768	8
Kin8nm	8192	8
Naval Propulsion Power Plant	11,934	16
Combined Cycle Power Plant	9568	4
Protein Structure	45,730	9
Red Wine Quality	1599	11
Yacht Hydrodynamics	308	6
Year Prediction MSD	515,345	90

**Table 2 entropy-23-00041-t002:** Mean NLL and standard error of the predictions. The entries in bold denote the best performing model(s).

Dataset	BLR	VI	FedAG${}_{N=0}^{\left(t=1\right)}$	FedAG${}_{N=1}^{\left(t=5\right)}$	DE
Boston	3.07±0.03	2.90±0.07	3.02±0.03	2.58±0.06	2.41±0.25
Concrete	3.78±0.02	3.39±0.02	3.76±0.03	3.21±0.04	3.06±0.18
Energy	5.12±0.05	2.39±0.03	5.31±0.06	2.07±0.04	1.38±0.22
Kin8nm	1.17±0.04	−0.90±0.01	1.03±0.04	−0.87±0.01	−1.20±0.02
Naval Propulsion	−3.55±0.02	−3.73±0.12	−3.45±0.01	−3.21±0.01	−5.63±0.05
Power Plant	2.97±0.01	2.89±0.01	2.94±0.01	2.92±0.01	2.79±0.04
Protein	3.07±0.00	2.99±0.01	3.08±0.00	2.95±0.00	2.83±0.04
Red Wine	1.50±0.07	0.98±0.01	1.01±0.03	0.99±0.02	0.94±0.12
Yacht	3.63±0.05	3.44±0.16	4.02±0.07	1.92±0.06	1.18±0.21
Year Prediction	3.73±NA	3.86±NA	3.72±NA	3.66±NA	3.35±NA

**Table 3 entropy-23-00041-t003:** RMSE and standard error of the predictions. The entries in bold denote the best performing model(s).

Dataset	BLR	VI	FedAG${}_{N=0}^{\left(t=1\right)}$	FedAG${}_{N=1}^{\left(t=5\right)}$	DE
Boston	4.87±0.22	4.32±0.29	4.96±0.22	4.07±0.18	3.28±1.00
Concrete	10.58±0.33	7.13±0.12	10.52±0.33	6.50±0.20	6.03±0.58
Energy	4.35±0.14	2.65±0.08	4.36±0.14	2.02±0.07	2.09±0.29
Kin8nm	0.20±0.00	0.10±0.00	0.20±0.00	0.10±0.00	0.09±0.00
Naval Propulsion	0.01±0.00	0.00±0.00	0.01±0.00	0.01±0.00	0.00±0.00
Power Plant	4.74±0.05	4.33±0.04	4.56±0.05	4.45±0.05	4.11±0.17
Protein	5.18±0.02	4.84±0.03	5.18±0.02	4.63±0.02	4.71±0.06
Red Wine	0.65±0.02	0.65±0.01	0.65±0.02	0.65±0.02	0.64±0.04
Yacht	9.12±0.52	6.89±0.67	9.12±0.52	2.29±0.15	1.58±0.48
Year Prediction	9.51±NA	9.03±NA	9.51±NA	9.35±NA	8.89±NA

**Table 4 entropy-23-00041-t004:** Mean determinant sharpness (DS) of the predictive distributions calculated with FedAG${}_{N=1}^{\left(t=5\right)}$ and DE.

Dataset	FedAG${}_{N=1}^{\left(t=5\right)}$	DE
Boston	4.05	4.79
Concrete	6.37	7.07
Energy	2.16	2.67
Kin8nm	0.10	0.14
Naval Propulsion	0.01	0.02
Power Plant	4.30	5.48
Protein	3.88	4.59
Red Wine	0.79	0.69
Yacht	2.85	0.94
Year Prediction	11.12	7.85

## Data Availability

Data sharing not applicable.

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
