# Peer review of "Probabilistic Predictions with Federated Learning"

_entropy, 2020, doi:10.3390/e23010041_

Round 1
Reviewer 1 Report
The authors present a study on probabilistic predictions using a federated learning approach where the set of local weights is interpreted as posterior distribution for the weights of a global model. The approach is interesting and relevant but somewhat limited in impact as only small NNs were evaluated. In addition, there are some issues with this manuscript that must be addressed:
- The state of the art is lacking some recent important contributions in probabilistic federated learning (e.g. Zhang et al., IEEE Probabilistic Solar Irradiation Forecasting based on Variational Bayesian Inference with Secure Federated Learning). Please include the relevant recent state of the art and describe your improvements.
- The methods are lacking important detail necessary to judge the validity of the approach, in particular how the data was divided over the devices. Stratified splits would probably perform better than non-stratified splits but the latter is more likely to occur in the real-world. It would be interesting to see a comparison of the approach with and without stratification but at least the approach taken should be explained clearly and put into context.
- Please also explain the architecture of the comparison NNs (i.e. VI and DE) and put it in context with your network designs and results achieved. For federated learning smaller networks may perform better as the available dataset for each learner is limited. Again, it would be nice to see a comparison of different network sizes to gain more insights into this but this could be left for future work.
- There are other hyperparameters in the manuscript that are not put in context well: The number of devices and the fraction of devices that will perform the client updates. Why did you choose the number of devices that you used and how would the results change with a different number? Why was the fraction C always chosen to be 1?
- Please also explain the rationale for setting the weights of the clients to the expectation values of the global model (line 15 in the pseudo code). Would it not be better to use this as another learning step with an update similar to line 18 (i.e. rather than setting the weights to the fixed value, update them based on their difference to the global model)? Also, could it be that you meant to say w <- E(p(w_t | D) rather than w_t+1 in line 15 of the pseudo code?
- In the results and discussion section, I am missing an interpretation of the results of Figure 3 (l 196 onwards).
- Likewise, a comparison of the computational complexity between DE and VI and the proposed approach is missing. In addition, a comparison of the communication complexity compared to FedAvg should be provided
- As a minor point: It would be helpful to repeat the citations of the comparison frameworks in line 156-157.
Reviewer 2 Report
Report on the paper \Probabilistic predictions with federated learning" by
Adam Thor Thorgeirsson and Frank Gauterin.
The paper is in the eld of machine learning (ML). The goal is to perform
probabilistic predictions of the predictive uncertainty type with the method
of federated learning (Fed). The chosen method is called FedAvg-Gaussian
(FedAG) built on the federated averaging (FedAvg). The new method is com-
pared to other methods such as Bayesian linear regression (BLR), variational
inference (VI) and deep ensembles (DE) all of which being implemented in a
non-distributed setting. The results in table 2 and 3 are with mean negative log-
likehood (NLL) and root mean square error (RMSE), respectively when applied
to some datasets available of the UCI machine learning repository [41].
Details of the new approach are in Sect. 3 where a pseudo-code is given.
The experimental validation is detailed in Sect. 4. To paraphrase the authors
\FedAg oers prediction performance comparable with state-of-the-art proba-
bilistic ML algorithms in an ecient and privacy-preserving manner".
I found the paper very readable although the number of acronyms is high
and dicult to digest by the non specialist. The work seems to add new matter
in the eld of ML. The paper is also well written and clean. I have no objection
for a publication in Entropy.
Author Response
Dear Prof. Knuth, dear Mr. Hindle, dear reviewers,
We appreciate the time and effort that you and the reviewers dedicated to providing feedback on our manuscript and are grateful for the insightful comments on and valuable improvements to our paper. We have incorporated the suggestions made by the reviewers. Those changes are highlighted in red within the manuscript.
Round 2
Reviewer 1 Report
The authors have addressed most of my comments and concerns. As a minor revision, it would be great to see a quantitative comparison for the computational complexity (similar to the comparison of the communication complexity). With this addition, the manuscript can be published.